# The Relationship Between Gut Microbiota, Muscle Mass and Physical Function in Older Individuals: A Systematic Review

**DOI:** 10.3390/nu17010081

**Published:** 2024-12-28

**Authors:** David J. Barry, Sam S. X. Wu, Matthew B. Cooke

**Affiliations:** 1School of Health Sciences, Swinburne University of Technology, Melbourne VIC 3122, Australia; sswu@swin.edu.au; 2Sport, Performance and Nutrition Research Group, School of Allied Health, Human Services and Sport, La Trobe University, Bundoora VIC 3086, Australia

**Keywords:** sarcopenia, physical performance, gut microbiota, gut–muscle axis, supplement, older adults

## Abstract

Background: Recent evidence suggests that sarcopenia and subsequent changes in muscle mass and functional outcomes are linked to disruption to the gastrointestinal microbiota composition and/or function via the microbiota-gut-muscle axis. Despite growing interest, few studies have systemically analysed (1) the relationship between the gut microbiota, muscle mass and physical performance and (2) the effects of gut-modulating dietary interventions on these outcomes within older individuals with or without sarcopenia. Methods: Four electronic databases (PubMed, MEDLINE, Embase and Scopus) were searched for articles published from the year 2004 until July 2023. The Preferred Reporting Items for Systematic Reviews and Meta-Analyses (PRISMA) were followed. Revised Cochrane Risk of Bias (RoB 2.0) and Joanna Briggs Institute (JBI) critical appraisal checklist were utilised to evaluate the risk of bias within intervention and observational studies, respectively. Results: A total of 20 studies (14 observational and 6 interventional) involving 4071 older participants (mean age 69.9 years, 51.6% female) were included. There was significant heterogeneity regarding interventions and outcome measures used in these studies. Correlations between microbiota diversity and composition and sarcopenia-related functional outcomes were observed. Interventional studies targeting the gut microbiota resulted in improved muscle strength, body composition or physical function in some, but not all, studies. Conclusions: Despite limitations in the studies reviewed, the findings provide further evidence that the development of sarcopenia is likely influenced by an altered gut microbial environment and that interventions targeting the microbiome could hold therapeutic potential for the treatment or management of sarcopenia.

## 1. Introduction

Aging is associated with declines in skeletal muscle mass and strength which can lead to impaired physical performance [1]. Sarcopenia, a progressive and generalised skeletal muscle disorder characterised by loss of lean muscle mass, strength and/or function [2,3], is concomitant with an increased risk of poor clinical outcomes including falls, disability, morbidity and mortality [4,5,6,7,8]. It has been well-established that the prevalence of sarcopenia increases with advancing age [9,10,11,12,13], although wide variations in prevalence rates have been reported, which may be due to heterogeneity of study populations [14] or a result of differences in operational criteria, assessment methods or cut-off values. Several international working groups have proposed consensus definitions for diagnosing sarcopenia; those from the European Working Group on Sarcopenia in Older People (EWGSOP2) and the Asian Working Group for Sarcopenia (AWGS) were recently updated [15,16]. While the cause of sarcopenia is multifactorial, emerging evidence suggests that the intestinal microbiome may play an important role [17]. However, whether it is the cause of the disease or simply a mediator, transforming environmental signals into physiological processes is debatable [18].

The human gastrointestinal tract (GIT) contains a diverse community of bacteria, fungi and viruses reported to be in the order of 10^14^ microorganisms. Anaerobic bacteria comprise the bulk of this complex ecosystem, predominantly represented by five bacterial phyla (Bacteroidetes, Firmicutes, Actinobacteria, Proteobacteria and Verrucomicrobia) [19]. There is an immense species-level diversity of GIT microbiota within human hosts [20,21] and high inter- and intra-personal variation have been observed [22,23]. The gut microbiota undergoes complex changes with advancing age, which can result in a different microbial profile compared to younger adults [24,25]. Reduced microbial diversity [26], increased variability between individuals, and an overgrowth of pathobionts [27] are some of these key differences. These changes, often described as “gut dysbiosis”, have been associated with inflammatory states, dysregulated immune responses, obesity, insulin resistance and type 2 diabetes [28,29,30,31]. However, lifestyle (especially diet), stress and medication use are all environmental factors that can contribute to dysbiosis, independent of aging. 

Interactions between the gut microbiota and skeletal muscle termed the microbiota–gut–muscle axis, have been an area of intense research activity, with several recent reviews highlighting the associations between age-associated changes in microbiota composition, skeletal muscle health and physical performance [18,32,33,34,35]. Studies suggest that altered microbial composition can promote anabolic resistance and persistent, low-grade systemic inflammation (“inflammaging”), resulting in reduced lean muscle mass, impaired muscle function and adverse clinical outcomes [17,36,37]. Reduced butyrate-producing bacteria such as *Clostridium* XIVa and *Roseburia* appear to be an important characterisation of the aging microbiome, which may link to muscle atrophy [38,39,40]. Microbiota-modulating supplements such as probiotics have shown to enhance muscle mass and strength in pre-clinical studies [41,42]. In addition, administration of butyrate to aged mice resulted in increased lean muscle mass [43].

There have been numerous reviews summarizing links between microbiota and muscle mass and function. However, at that time, the majority of evidence came from animal models or observational studies, with limited interventional human studies published, especially in older adults with or without sarcopenia. Further, despite several systematic reviews (one with meta-analysis) also published on this topic [44,45,46,47], no systematic review has included gut-modulating interventions to explore potential cause-and-effect relationships. Therefore, this systematic review aims to examine the relationship between the human gut microbiota, including modulators of the gut microbiota on muscle strength, muscle mass and/or physical performance in older populations.

## 2. Materials and Methods

This systematic review was conducted following the Preferred Reporting Items for Systematic Reviews and Meta-Analyses (PRISMA) guidelines [48]. The protocol was registered in the International Prospective Register of Systematic Reviews (PROSPERO) (CRD: 42021273717). A subsequent review was conducted in July 2023 and the PROSPERO record was revised.

### 2.1. Search Strategy

We performed a search of NIH/PubMed, MEDLINE (Ovid), Embase and Scopus using the following search string: (“microbiome” OR “microbiota” OR “gut microbiome” OR “gastrointestinal microbiome”) AND (“elderly” OR “aged” OR “aging” OR “ageing” OR “older persons”) AND (“sarcopenia” OR “skeletal muscle” OR “muscle mass” OR muscle strength” OR “muscle atrophy” OR “grip strength” OR “gait speed” OR “physical performance” OR “muscle quality” OR “lean mass” OR “frailty”) AND (“prebiotic” OR “probiotic” OR “synbiotic” OR “symbiotic”). Exploded MeSH or thesaurus terms were used where possible. Studies were imported into a reference manager (Endnote X9™) for deduplication.

The above keyword search terms were used on two separate occasions as below:

Search 1 (20 September 2021): for the period between January 2004 and September 2021. This time interval reflects next-generation sequencing techniques for gut microbiota.

Search 2 (3 July 2023): for the period between September 2021 and July 2023.

Therefore, all articles published between January 2004 and July 2023 were considered. Grey literature sources were not searched. Reference lists of selected studies were also searched manually.

### 2.2. Study Selection

Duplicate articles returned from the search were removed (DB). Titles and abstracts of all returned citations were then independently screened for eligibility by two reviewers (DB and MC). The same reviewers then screened the relevant full-text articles for inclusion based on our inclusion/exclusion criteria. Disagreements were resolved by discussion with a third reviewer (SW). A list of excluded papers and explanations for exclusion can be found in Appendix A.

### 2.3. Inclusion and Exclusion Criteria

Observational, cross-sectional, longitudinal and interventional study designs published in English were considered. Articles were deemed eligible if: (1) study populations were older adults (defined as mean age >50 years); (2) assessed outcomes that included at least one of muscle strength (e.g., grip strength), muscle mass (e.g., appendicular lean mass) or physical performance (e.g., gait speed), in alignment with a clinical diagnosis of sarcopenia and (3) clinical trials included relevant baseline measurements. Editorials, commentary articles, letters to the editor, theses, dissertations and book chapters were excluded. Trials that did not distinguish treatment and control groups or interventions using combined exercise components were excluded. Figure 1 depicts the PRISMA flow diagram for this systematic review.

### 2.4. Data Collection and Extraction

Two reviewers (DB and MC) extracted data independently using the Cochrane Collaboration template [49]. Data extracted from all studies included author(s), publication year and country, study design, sample characteristics (number, mean age and gender when available) and key findings. Methods for assessment of dietary habits and physical activity, specific outcome measures (how muscle mass, strength and physical performance were assessed), and gut microbiota analysis (e.g., DNA extraction and sequencing) were included for synthesis and comparison of findings. In addition to the above, data extracted from interventional studies included characteristics of intervention (species/strains, dose, duration) and comparator (e.g., placebo) and results were mapped to an intervention. No assumptions were made on missing or unclear data. In the event of missing and/or unclear data, the authors of the article were contacted in an attempt to provide this information. In cases where this contact was unsuccessful, the article was excluded. Discrepancies were resolved by a third reviewer (SW).

### 2.5. Data Synthesis and Analysis

Endnote X9™ was used for synthesis and collation of data. Eligible papers were grouped into (i) observational (cross-section and case–control) and (ii) interventional (parallel and cross-over) studies. Due to the heterogeneity of included studies, a meta-analysis of outcomes was not possible. Instead, a qualitative review was performed.

### 2.6. Assessment of Methodological Quality

Two independent reviewers conducted a quality appraisal of the included studies using the revised Cochrane Risk of Bias 2 (RoB 2) [50] and critical appraisal tools from the Joanna Briggs Institute (JBI) [51]. Bias in observational studies was assessed using JBI critical appraisal tools for analytical cross-sectional and case–control studies (DB and MC). The methodological quality of interventional trials was analysed according to the RoB 2 framework for both parallel and cross-over design studies (DB and SW).

## 3. Results

A total of 784 studies were retrieved through electronic database search (Figure 1). After removing duplicates (*n* = 150), 634 records were identified for screening. Once studies were screened by title and abstract, 534 were excluded, resulting in 100 articles for full-text review. One additional paper was identified after reviewing the reference lists of these studies. These articles were screened fully to assess their eligibility. A total of 20 studies satisfied all inclusion and exclusion criteria, including 6 interventional trials [52,53,54,55,56,57] and 14 observational studies [36,58,59,60,61,62,63,64,65,66,67,68,69,70].

### 3.1. Characteristics of Reviewed Studies

Eight studies were conducted in Asia [54,61,62,65,67,68,69,70], six in Europe [36,52,57,59,63,64], four in the USA [55,56,58,60], one in Canada [66] and one in Brazil [53]. Two studies were conducted on women only [55,68] and one study recruited men only [58]. Fifteen studies recruited both women and men [36,52,53,54,56,60,61,62,63,64,65,66,67,69,70] while two studies did not report the biological sex of participants [57,59]. In total, 4071 participants were included, of which approximately 51.6% were female. The mean age of participants across all 20 studies was 69.9 ± 8.3 with sample size ranging from 17 to 1417 participants. General characteristics of the included studies are outlined in Table 1.

#### 3.1.1. Observational Studies

There were 14 observational studies, including seven secondary analyses of previous studies [58,59,60,63,64,67,70]. Sample sizes ranged from 17 to 1417, and a total of 3825 participants were included (69.0 ± 6.4 years, approximately 60.1% female). One study did not report the biological sex [59]. One study enrolled only males [58], one recruited only females [68], and the remaining 11 studies included both males and females. Table 2 summarises the characteristics and main findings of observational studies included in this review. Please note that only healthy individuals, including those with sarcopenia and/or frailty, were analysed. In those studies that included a disease such as HIV or overweight individuals as a comparison, only the healthy control group was analysed.

#### 3.1.2. Interventional Studies

This review identified six interventional trials. The total number of participants was 246 and sample sizes ranged from 17 to 66 (approximately 55.3% female). One study included females only [55], one study did not report gender [57] and the remaining four studies recruited both females and males [52,53,56,71]. The mean age across all six trials was 73.4 ± 4.8 and the duration of intervention ranged from 2 months to 18 weeks. A summary of the characteristics and main findings of interventional studies included in this review is described in Table 3.

None of the study interventions used were directly comparable. One study used a single-strain (*Lactobacillus plantarum*) probiotic [71]. One study used a commercially available prebiotic containing fructooligosaccharides (FOS) plus inulin [52]. A multi-strain synbiotic was used in two studies: one provided *Lactobacillus paracasei*, *Lactobacillus rhamnosus*, *Lactobacillus acidophilus* and *Bifidobacterium lactis* with FOS [53], while *Bifidobacterium bifidum*, *Bifidobacterium breve*, *Bifidobacterium longum*, *Lactobacillus acidophilus* and *Lactobacillus plantarum* plus inulin was utilized in the other [55]. One study [56] used a postbiotic (urolithin A), while an experimental formula containing *Lactobacillus paracasei*, omega-3 fatty acids, and leucine was used in another [57]. A placebo comparator was used in 100% of the studies and was most commonly (4/6) maltodextrin. See Table 3 for more details.

### 3.2. Assessment of Muscle Strength

Muscle strength was evaluated in 18 studies. Hand grip strength (HGS) was utilised in sixteen studies [36,52,53,55,57,58,60,61,62,64,65,66,67,68,69,71], six studies used repeat chair stands (RCS) [36,59,60,61,67,71], and both were measured in five studies [36,60,61,67,71]. Table 4 provides a summary of the methods used for evaluating muscle strength, mass and physical performance.

### 3.3. Assessment of Muscle Mass

Seventeen studies evaluated muscle mass. Bioelectric impedance analysis (BIA) [36,53,55,61,65,67,68,69,70] and dual-energy X-ray absorptiometry (DXA) [57,58,59,60,63,64,71,72] were the most common methods utilised in nine and eight studies, respectively, while magnetic resonance imaging (MRI) was used in one study [56]. Muscle mass (quantity) was reported in a variety of ways, including absolute values such as fat mass (FM), fat-free mass (FFM), lean body mass (LBM), skeletal muscle mass (SMM), appendicular skeletal muscle mass (ASMM) or appendicular lean mass (ALM); indices such as skeletal muscle index (SMI) or appendicular skeletal muscle index (ASMI); or by incorporating adjustments to account for body size (height, weight, BMI) such as ALM/ht^2^, LBM/ht^2^, ASM/body weight, ALM/BMI.

### 3.4. Assessment of Physical Performance

Among the 13 studies that evaluated physical performance, the short physical performance battery (SPPB) was performed in seven studies [36,57,58,60,63,67,69]. Gait speed (GS) was assessed in six studies, reporting walking time across 4- [36,52,65], 6- [67], and 10 m distances [71]. GS was evaluated in the study by Wang et al. [68]; however, the distance was not specified. Six-minute walking distance (6MWD) was measured in two studies [56,64], and the timed up and go (TUG) test was evaluated in one [71]. Dillon et al. [60] evaluated the 400 m walking time and stair climb test.

### 3.5. Assessment of Gut Microbiota/Microbiome

Taxonomic composition profiling of faecal samples was undertaken in 14 studies. Faecal metagenomic sequencing was utilised to quantify the microbiota in two studies [36,67], while 12 studies performed 16S rRNA sequencing, amplifying hypervariable regions V4 [55,58,59] and V3–V4 [60,61,62,63,64,65,66,69,70]. Various alpha-diversity indices were used to evaluate differences in the richness and evenness of samples in 11 studies [58,59,60,61,62,63,64,65,66,67,69] and variability in the community composition across samples (beta-diversity) was reported in 10 studies [36,58,59,61,62,64,65,66,67,68]. Gut microbial functions were evaluated by predictive metagenome profiling (PICRUSt) in two studies [61,62]. A summary of methods for evaluating gut microbiota and microbiome is provided in Table 5.

### 3.6. Risk of Bias and Quality Appraisal Assessment

Quality assessment using JBI critical appraisal tools for analytical cross-sectional and case–control studies are displayed in Figure 2 and Figure 3, respectively. The risk of bias for parallel and cross-over design interventional studies using RoB 2 is displayed in Figure 4 and Figure 5. Of the observational studies, the majority addressed the criteria and thus were rated quite high. However, only 3/6 studies addressed the criteria of identifying and controlling for confounding factors. This was similar for the case–control studies, with 4/8 studies not identifying and controlling for confounding factors. Moreover, measurement of exposure between case and control was not identified in 4/8 studies. For the parallel study designs, the majority met all criteria, and the risk of outcome assessment was mostly low. One study [52] raised some concerns regarding the reporting of selected data. In addition, 3/5 studies were funded by the supplement/product company and listed co-authors that worked or have been funded before by said supplement/product company [53,56,57]. Of the six interventional studies, only three registered their trial with their respective clinical trials registry [52,53,54]. The one cross-over study [55] was rated overall high risk, specifically regarding carry-over effects, selection of reported data and study funded by the supplement/product company and listed co-authors working or have been funded before by said supplement/product company.

## 4. Discussion

As human life expectancy increases, age-related diseases, particularly sarcopenia, are becoming more prevalent, posing a significant global health burden. The emergence of the gut microbiota and its potential influence on muscle health via the microbiota-gut–muscle axis has become an area of increased scientific interest, but also a target for dietary manipulation to mitigate sarcopenia and/or enhance muscle health. From the 14 studies included in the current review that explored the associations between microbiota diversity and/or composition and assessments of muscle mass and performance, a total of 6 studies compared directly sarcopenic to non-sarcopenic individuals, while the remaining used relatively healthy community-dwelling older individuals. Findings from both small and large (population) studies included demonstrated clear microbiome differences between sarcopenic and non-sarcopenic individuals. Several key bacterial taxa, especially those linked to butyrate production, were found to be of significant relevance when linking altered human gut microbial composition and impaired muscle strength, mass and/or function. Intervention studies that targeted the microbiota failed to show benefit on muscle mass, function and/or indices of sarcopenia. However, results should be interpreted with caution due to their limitations or low study quality. Taken together, reduced microbial diversity and/or taxonomic composition differences between healthy older and sarcopenic individuals suggest the presence of a distinct microbiome, which we conceptually refer to as the “sarcobiome”. More clinical (nutrition) trials are needed to confirm the purported benefits of dietary interventions targeting the gut microbiota to improve muscle health and/or mitigate the pathogenesis of sarcopenia.

### 4.1. Summary of Main Results

In one of the larger population studies, secondary analysis revealed significant differences in beta diversity (Bray–Curtis distance) between individuals with and without sarcopenia [67]. The authors reported that several bacterial species, including *D. piger*, *C. symbiosum*, *H. effluvii*, *B. fluxus*, *C. innocuum*, *C.* and *C. citroniae*, were significantly associated with the presence of sarcopenia and its severity [67]. The same authors conducted a smaller case–control study but only in women and found nine distinct bacterial species were significantly enriched in sarcopenic versus 13 species in the non-sarcopenic individuals, including associations with muscle mass [68]. From these studies, the relative abundance of *D. piger*, *P. copri* and *B. longum*, were considered potential taxonomic biomarkers of sarcopenia. However, with low to moderate AUC values for *P. copri* (AUC: 0.372) and *B. longum* (AUC: 0.647), only *D. Piger* was identified as most important bacteria for discriminating between sarcopenia and non-sarcopenic (AUC: 0.852). The ratio Firmicutes/Bacteroidetes (F/B) could also be considered a biomarker of sarcopenia, with Wang and colleagues reporting a higher F/B ratio in sarcopenic individuals compared to controls [68]. The F/B ratio is widely considered to be important for maintaining normal intestinal homeostasis, and a decrease in the ratio is often observed in inflammatory diseases [73]. Similarly, Wu and colleagues found the ratio of Prevotella/Bacteroidetes (P/B) to be higher (1.5 times) in sarcopenic compared to non-sarcopenic individuals [69]. In addition, the genus *Coprococcus* was positively associated with sarcopenia, while the family Lachnospiraceae demonstrated a negative relationship. The P/B ratio is considered an important indicator of diet and lifestyle, and like the F/B ratio, it may also be a possible biomarker for a distinction between control and sarcopenia individuals [74]. In another cross-sectional study, higher levels (5 times) of *F. prausnitzii* were evident in non-sarcopenic individuals compared to individuals with primary sarcopenia and these levels were significantly associated with higher hand grip strength [66]. Relative abundances of *F. prausnitzii* and *B. longum* are likely beneficial species for healthy aging and muscle, given their role in short-chain fatty acid (SCFA) production. In addition, *F. prausnitzii* is associated with improved insulin sensitivity, anabolic balance and alleviation of local intestinal and systemic inflammation [26,75,76]; all beneficial to healthy muscle mass and function.

The addition of physical frailty to the microbiome–sarcopenia relationship was recently explored by Picca and colleagues [77]. Biomarkers of physical frailty appeared to be identified along with sarcopenia using machine classifier techniques. Among the gut microbes contributing to the model, *Oscillospira* and *Ruminococcus* were more abundant in such individuals, while *Barnesiellaceae* and *Christensenellaceae* were higher in the non-physically frail/sarcopenic. While very little is known about a possible functional link between *Oscillospira* and physical frailty and sarcopenia, the butyrate-producing bacteria *Ruminococcus* is in line with previously reported associations between *Ruminococcus* abundance and frailty [33]. Indeed, a secondary analysis by Tavella and colleagues [64] found muscle mass (SMI) to be negatively correlated with *Ruminococcus 2* along with *Blautia*, *Fusicatenibacter*, and *Subdoligranulum*, whereas SMI was positively correlated with the *Christensenellaceae* R7 group. In another study, SMI was significantly positively correlated with *Blautia* and *Bifidobacterium* in men, while in women, higher SMI was associated with higher read counts of *Eisenbergiella*, *Flavonifractor*, *Eggerthella*, and *Erysipelotrichaceae incertae sedis* [70]. Whether and how Ruminococcus abundance impacts muscle metabolism and function in the context of sarcopenia and/or frailty warrants further investigation. Finally, Lim et al. [62] reported a significant association between interindividual variation in microbiota composition (Bray–Curtis distance) and frailty-related metadata, including grip strength and the Korean Frailty Index (FI) score. Grip strength was not significantly associated with various alpha diversity indices, although a weak association was observed with *Bacteroides* and *Alistipes.*

Based on the observations in older sarcopenic and/or frail individuals, there appears to be an overall apparent shift of gut microbiota composition towards a lower abundance of butyrate-producing bacteria with physical frailty and sarcopenia, suggesting a positive role for these microbes in muscle function and skeletal mass potentially via direct action on the muscle or secondary action via reductions in inflammation. Indeed, Kang and colleagues showed a decrease in the abundance of butyrate-producing genera (*Lachnospira*, *Fusicantenibacter*, *Roseburia*, *Eubacterium* and *Lachnoclostridium*) in sarcopenic and “probable” sarcopenic subjects [61]. The apparent protective effect of butyrate-producing bacteria on outcomes related to sarcopenia suggests a potential benefit of enhancing the colonisation and/or growth of these bacteria through diet or probiotic supplementation.

With this in mind, Barger et al. found that FFQ-reported diets high in fibre positively impacted butyrate-producing genera (including *Ruminococcus*, *Lachnospira* and *Clostridia*), and gene counts for butyrate production (KEGG IDs: K01034, K01035), with authors suggesting a likely mechanism of action that may link dietary fibre intake with higher levels of whole-body lean mass (%) and physical function in older adult men [58]. Cox and colleagues underwent a secondary analysis of older female individuals to explore differences in microbiota and muscle mass/strength in those with poor appetite (Simplified Nutritional Appetite Questionnaire [SNAQ] score < 14) compared with normal appetite [59]. Differences in bacteria species involved in producing butyrate were evident between those with good and poor appetite, with reductions in *Lachnospira* as well as significantly higher mean chair stand time (indicating lower muscle strength) observed in those with poor appetite [59]. However, this was based on within-group observations only and no direct correlation between relative bacterial abundance and performance measures was undertaken. Notwithstanding, both poor and good appetite groups were matched on dietary similarity via a healthy eating index, which may suggest an influence of appetite on bacteria abundance beyond dietary intake alone [78]. Conversely, it may also reflect a limitation of using self-report mechanisms for capturing dietary data [79].

Finally, in two separate studies that examined faecal butyrate levels with relatively healthy 60–70 year olds, faecal butyrate levels were found to be positively correlated with sex-adjusted SMI [65] and greater grip strength [60]. These observations add further support to the mechanisms of the microbiota-gut-muscle axis via the production of SCFA butyrate; however, it should be noted that only 5% of all microbially produced SCFAs can be found in feces and thus other measures (i.e., microbiome and blood) may be required to confirm such a relationship [80].

Few studies have explored the effects of dietary (gut-modulating) interventions on muscle mass and function in healthy and/or sarcopenic individuals. The studies included in the review used single- and multi-strain probiotics with or without prebiotics or combined with other ingredients. Two studies investigated the effects of multi-strain synbiotic formulations on muscle strength and/or lean muscle mass. Treatment for 3 months with a synbiotic (6 g FOS + *L. paracasei*, *L. rhamnosus*, *L. acidophilus* and *B. lactis*, all at 10^8^ to 10^9^ CFU) resulted in no difference in HGS or FFM compared with placebo in older male and females [53]. Conversely, an 18-week crossover trial in older women given a high protein maintenance diet combined with either a prebiotic (inulin), a multi-strain (1.54 × 10^9^ CFUs *B. bifidum* and 4.62 × 10^9^ CFUs each of *B. breve*, *B. longum*, *L. acidophilus*, and *L. plantarum*) probiotic, or synbiotic (prebiotic + probiotic) resulted in a small, but significant increase in FFM [55]. This study also analysed the microbiota, and while bacterial diversity was not significantly impacted, *Lactococcus* and *Streptococcus* were enhanced, while butyrate-producing genera *Roseburia* and *Anaerostipes* were suppressed [55]. Several limitations existed in this study, including a short intervention period (2 weeks), carry-over effects (confirmed by faecal qPCR recovery) due to cross-over design and other dietary factors (increasing protein intake during the study)—all potentially influencing the results and thus should be interpreted with caution.

In another study, patients with sarcopenia consuming an experimental formula (*L. paracasei*, leucine and omega-3 fatty acids) for 60 days demonstrated significant improvements in HGS and SPPB scores [57], however, because the intervention was a multi-ingredient product, it is difficult to ascertain which ingredient or ingredients led to the improvements [57]. Performance improvements were also observed after consuming a single-strain probiotic formulation containing *L. plantarum* for 18 weeks [54]. Significantly lower TUG times, but not 10 m walking test, were noted in the intervention compared to the placebo group [54]. Again, limitations existed within the study, such as limited recording of the actual amounts of food intake and physical activity during the study, which may impact the outcomes. While positive effects were observed, given no microbiome analysis, it is difficult to ascertain possible mechanisms of action. Finally, Liu et al. reported a non-significant improvement in 6MWD following daily oral supplementation of a postbiotic (1000 mg urolithin A) compared with placebo [56]. Urolithin is a gut microbiome-derived metabolite that has been shown to stimulate mitophagy, improve muscle function in older animals and induce mitochondrial gene expression in older humans.

### 4.2. Limitations

Several limitations exist within the studies included: Firstly, there is substantial variation regarding the setting (e.g., sample size, participant health status or co-morbidity, and geographic location) and the methodology of the studies included in this review. Several critical methodologies differ from one study to another. For example, faecal sample collection and processing, the platform used for sequencing, and sequencing depth can all impact results [81,82]. Interestingly, the time of day for stool sample collection appears to be associated with the gut microbiome. A small (*n* = 6) cross-sectional study of healthy female volunteers who defecated more than once daily reported the second stool of the day had lower bacterial richness and diversity and significantly higher total SCFA concentrations compared with the first stool [82]. Habitual diet and physical activity (or their potential changes during the intervention period for experimental trials) did not appear to be adequately assessed or controlled in multiple studies. Therefore, the observed gut microbiota outcomes may have been impacted by these lifestyle factors, and it is not possible to draw firm conclusions about these interactions. Many different methods were used for evaluating the strength, muscle mass, and/or performance of the study populations, which limits the comparability of the outcomes. Moreover, in the case of using BIA and DXA, the accuracy and reliability of results may differ between methods and formulas used by the respective techniques. Although not critical to our review, the specific criteria that were followed for detecting or confirming sarcopenia were reported in some (9/20), but not all studies. Perhaps most importantly, in all described clinical trials, the dose, duration and intervention used differed from each other. Therefore, definitive conclusions about their effectiveness cannot be extrapolated. Additionally, a lack of longitudinal studies, particularly with post-intervention follow-ups, restricts understanding of both effect and effect duration. Finally, over half (4/6) of the interventional studies did not evaluate any changes in microbial composition or diversity following treatment, so attempting to determine mechanisms of action or causal relationships is limited.

### 4.3. Recommendations and Future Directions

This review identified a lack of longitudinal studies investigating the effects of targeted interventions on gut microbiota and/or its microbiome in older populations. Further research is needed in the form of controlled clinical trials, including different types of gut-modulating therapeutics, larger sample sizes of more homogenous populations, and extended follow-up periods. Ideally, these would be of adequate dose and duration and be designed as randomised, double-blinded, and placebo-controlled to avoid complications such as inadequate wash-out that may be experienced with cross-over study designs [83]. Future studies should ensure they include measures for capturing detailed information on diet composition, physical activity, and other factors that influence gut microbiota, such as sleep patterns and stress levels [84,85]. Insofar as possible, we recommend utilising direct methods (e.g., accelerometry) in addition to self-reported questionnaires to measure physical activity [86]. Likewise, methods that provide for both quantitative and qualitative analysis (such as 3–5-day food records or multiple 24 h dietary recalls) should be used for dietary assessment, leveraging digital platforms when practical, to minimise burden, maintain compliance and optimise data quality [87,88]. It would also be relevant to document medication use and consumption history of prebiotic- or probiotic-containing supplements. Identification and sub-analysis of baseline gut microbiota may help explain interindividual differences following treatment (“responders” and “non-responders”) and give valuable insight into underlying mechanisms of action. Adopting universal and standard methods for evaluating skeletal muscle strength, mass, and physical performance would better allow the combining of data for statistical modelling.

## 5. Conclusions

The findings of this review expand on previously published reports examining the associations between gut microbiome and muscle mass and physical function in older individuals with or without sarcopenia. Despite limitations in the reviewed studies, especially within the intervention studies, the available evidence suggests the presence of a “sarcobiome” within sarcopenic with or without frailty compared to non-sarcopenic, healthy older adults. Drawing substantive conclusions on the effects of prebiotics and/or probiotics on the gut microbiota is limited due to the heterogeneity of the interventions and research outcomes, study limitations and/or low-quality study designs. Notwithstanding, the utilisation of strategic microbiota-modulating therapeutics that target the key characteristics of the “sarcobiome” linked to muscle health could be a promising strategy for improving disease progression and symptoms. Additional longitudinal studies with a clear assessment of confounding variables and consistency of interventions are needed to better elucidate the causal relationships between deficits in skeletal muscle strength, mass and physical performance observed with aging and changes in the gut microbiome and to evaluate the effects of microbiota-modulating interventions.

## Figures and Tables

**Figure 1 nutrients-17-00081-f001:**
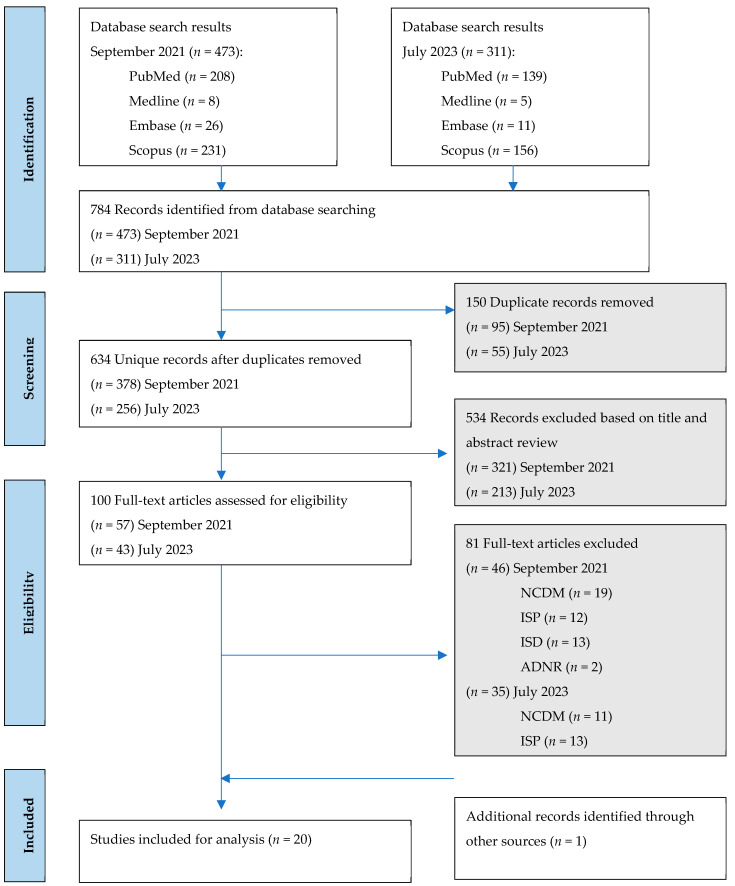
PRISMA flow diagram. ADNR = additional data not received; ISD = incorrect study design; ISP = incorrect study population; NCDM = no clear description of measurement.

**Figure 2 nutrients-17-00081-f002:**
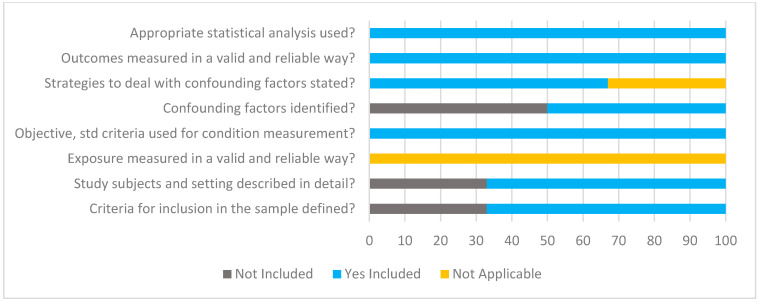
Summary of JBI critical appraisal for analytical cross-sectional studies. *n* = 6 studies were included in the critical appraisal for analytical cross-sectional studies. Please note the checklist descriptions have been modified slightly to fit within the figure.

**Figure 3 nutrients-17-00081-f003:**
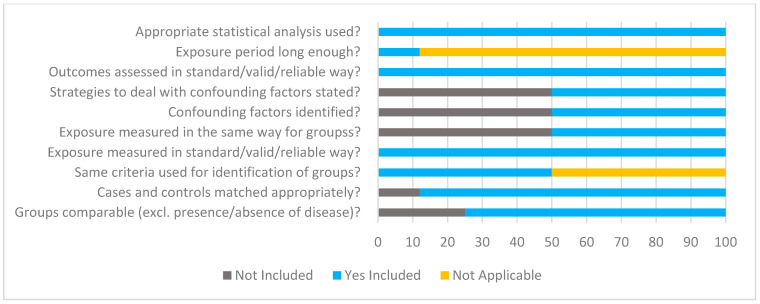
Summary of JBI critical appraisal for case–control studies. *n* = 8 studies were included in the critical appraisal for case–control studies. Please note the checklist descriptions have been modified slightly to fit within the figure.

**Figure 4 nutrients-17-00081-f004:**
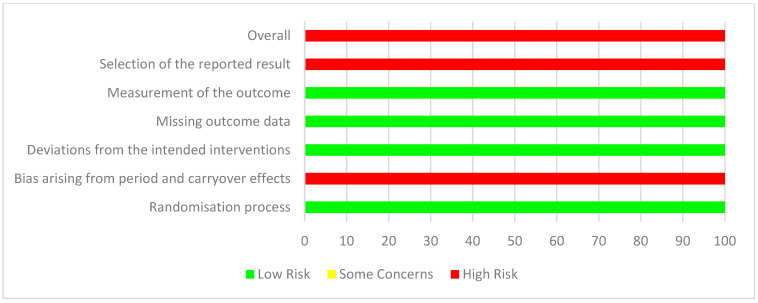
Summary of RoB 2 risk of bias for cross-over design interventional studies. *n* = 1 study was included in the analysis of risk of bias for cross-over design interventional studies.

**Figure 5 nutrients-17-00081-f005:**
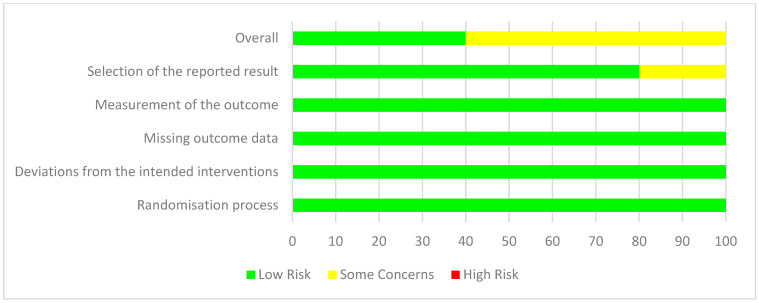
Summary of RoB 2 risk of bias for parallel design interventional studies. *n* = 5 studies were included in the analysis of risk of bias for parallel design interventional studies.

**Table 1 nutrients-17-00081-t001:** General characteristics of included studies.

Year	Authors	Country	Study Design	Population	Size (*n*) *
2020	Barger et al. [58]	USA	Secondary Data Analysis	Community Dwelling	74
2016	Buigues et al. [52]	Spain	RDBPCT	Nursing Home Residents	50
2021	Cox et al. [59]	UK	Cross-Sectional	Community Dwelling	204
2021	Dillon et al. [60]	USA	Cross-Sectional	Community Dwelling	36
2020	Ford et al. [55]	USA	Crossover RDBPCT	Community Dwelling	21
2022	Han et al. [65]	Taiwan	Cross-Sectional	Community Dwelling	88
2021	Kang et al. [61]	China	Prospective Pilot	Community Dwelling	87
2021	Lee et al. [54]	Taiwan	RDBPCT	Nursing Home Residents	42
2021	Lim et al. [62]	Republic of Korea	Cross-Sectional	Community Dwelling	176
2022	Liu et al. [56]	USA	RDBPCT	Community Dwelling	66
2013	Neto et al. [53]	Brazil	Randomised, Double-Blind Pilot	Community Dwelling	17
2019	Picca et al. [63]	Italy	Cross-Sectional	Community Dwelling	35
2022	Rondanelli et al. [57]	Italy	RDBPCT	Elderly with Sarcopenia	50
2023	Shah et al. [66]	Canada	Retrospective, Observational	Community Dwelling	350
2022	Sugimura et al. [70]	Japan	Cross-Sectional	Community Dwelling	848
2021	Tavella et al. [64]	Italy	Cross-Sectional	Community Dwelling	201
2020	Ticinesi et al. [36]	Italy	Cross-Sectional	Community Dwelling	17
2022	Wang et al. [67]	China	Cross-Sectional	Community Dwelling	1417
2023	Wang et al. [68]	China	Case–Control	Community Dwelling	100
2022	Wu et al. [69]	China	Case–Control	Elderly Hospitalised Patients and Healthy Volunteers	192

Note: * sample size represents the number of participants who completed the study or for whom complete datasets are available. RDBPCT: randomised, double-blind, placebo-controlled trial.

**Table 2 nutrients-17-00081-t002:** Characteristics and main findings of observational studies.

Study	Participants * (Age Mean ± SD)	Gender (M/F)	Study Design	Direct and/or Indirect Links Between Muscle Strength, Mass or Physical Performance and Gut Microbiota
Barger et al. (2020) [58]	*n* = 74; T3T3 *n* = 42 (84.4 ± 4.3); T1T1 *n* = 32 (84.4 ± 4.4) T3T3: Highest Tertiles for Dietary Fibre Density and %WBLM; T1T1: Lowest Tertiles.	74/0	Secondary Data Analysis of MrOS Study	Highest tertile for dietary fibre intake (and highest tertile for the percentage of whole-body lean mass) dis-played significantly higher lean mass % and higher values for SPPB and grip strength. Significant differences for β-diversity were identified in T3T3 Versus T1T1, specifically, butyrate-producing bacteria (*Ruminococcus*, *Lachnospira and Clostridia*).
Cox et al. (2021) [59]	*n* = 204 (67); Case *n* = 102 (68.0); Control *n* = 102 (67.6) Case: SNAQ Score < 14; Control: SNAQ Score > 14	Both Case and Control were 95.1% F	Secondary Data Analysis of Twins UK Cohort	Older individuals with reduced appetite demonstrated lower RA of species *Lachnospira pectinoschiza*, and *Lachnospira*, Bacteroides, and *Lachnospiraceae* UCG-004 at genus level. These individuals also displayed reduced muscle strength compared to those with “Good” appetite.
Dillon et al. (2021) [60]	*n* = 36 (51–74); PWH *n* = 14 (61.0 ± 7.7); CON *n* = 22 (59.5 ± 6.6) PWH: Person with HIV CON: Healthy Controls	35/1	Secondary Data Analysis of Exercise in Healthy Aging and Assessing Tenofovir Pharmacology in Older HIV-Infected Individuals Studies	Higher RA of *Bacteroides*, *Alistipes*, *Bacteroides*, *and Sutterella* was significantly associated with greater muscle strength. Higher RA of *Collinsella* and *Bifidobacterium* were significantly associated with lower muscle strength. Higher RA of *Bifidobacteria* was associated with a longer time to complete a stair climb or 400-m walk. Higher RA of *Catenibacterium* and *Butyrivibrio* were associated with greater estimates of LBM and ALM. increasing levels of butyrate and propionate were significantly associated with greater grip gtrength.
Han et al. (2022) [65]	*n* = 88 (>65) NM *n* = 52 (70.0 ± 4.2) LM *n* = 36 (72.3 ± 5.4) NM: Normal Muscle Mass LM: Low Muscle Mass	28/60 NM: 20/32 LM: 8/28	Cross-Sectional	The LM group had lower HGS compared with the NM group. ⍺-diversity was significantly lower in the LM group, and β-diversity was significantly different between NM and LM. RA of firmicutes and firmicutes/bacteroidetes ratio were significantly lower in LM group. RA of *Flavonifractor* was higher while RA of *Marvinbryantia*, *Ruminococcaceae* and *Akkermansia* were lower in the LM group. *Akkermansia* sp., was positively associated with grip strength, whereas two potentially pro-inflammatory gut bacteria, *Ruminococcus gnavus* and *Fusobacterium* sp., were inversely correlated with gait speed. LM group displayed reduced faecal butyrate. Faecal butyrate positively correlated to the sex-adjusted SMI.
Kang et al. (2021) [61]	*n* = 87; Case *n* = 11 (74.7 ± 8.6); preCase *n* = 16 (74.0 ± 6.9); CON *n* = 60 (68.4 ± 5.8) Case: Sarcopenia; preCase: Possible Sarcopenia; CON: Healthy Controls	36/51	Prospective Pilot Study	Indicators of species richness (Choa1) was significantly higher in control compared to preCase/Case. β-diversity was different between control and preCase/Case. RA of *Lachnospira*, *Fusicantenibacter*, *Roseburia*, *Eubacterium*, and *Lachnoclostridium* were significantly reduced in the Case and preCase, whereas *Lactobacillus* was significantly more abundant. A significant correlation was identified between ASMI and *Roseburia* and *Eubacterium*, indicating These genera are less abundant in individuals with decreased muscle mass. HGS was positively correlated with *Eubacterium* and *Lachnospira*, while RCS was negatively correlated with *Roseburia*, *Eubacterium* and *Lachnospira*.
Lim et al. (2021) [62]	*n* = 176 (74.7 ± 4.4)	54/122	Cross-Sectional	Weak negative associations between HGS and RA of *Bacteroides* and *Alistipes*. Lower RA of *Prevotella copri* and *Coprococcus Eutactus* and higher RA of *Bacteroides fragilis* and *Clostridium hathewayi* were observed in frail individuals.
Picca et al. (2019) [63]	*n* = 35 PF&S *n* = 18 (75.5 ± 3.9) nonPF&S *n* = 17 (73.9 ± 3.2) PF&S: Physical Frailty and Sarcopenia; nonPF&S: Nonsarcopenic, Nonfrail Controls	20/15	Secondary Data Analysis of BIOSPHERE and Gut Liver Axis Studies	No difference in ⍺-diversity between PF&S and nonPF&S. Higher RA of *Pyramidobacter*, *Dialister*, *Eggerthella;* and lower RA of *Slackia* and *Eubacterium* Identified in PF&S participants compared with nonPF&S. Machine classifier biomarker modelling revealed *Oscillospira* and *Ruminococcus* to be most abundant in PF&S participants, while *Barnesiellaceae* and *Christensenellaceae* were higher in the nonPF&S group.
Shah et al. (2023) [66]	*n* = 350 Normal *n* = 267 (56.2 ± 6.4) Overweight *n* = 83 (57.9 ± 6.2)	86/264	Retrospective, Observational	Higher RA of *Faecalibacterium Prausnitzii* in males with normal BMI and high HGS, and overweight females with high HGS **.
Sugimura et al. (2022) [70]	*n* = 848 Men *n* = 353 (50.0 ± 12.9) Women *n* = 495 (50.8 ± 12.8)	353/495	Secondary Analysis of Iwaki Health Promotion Project	Higher RA of *Eisenbergiella* and *Bifidobacterium* was associated with a greater ASM/BW in women and men, respectively. Lower RA of *Dorea* in women was associated with lower ASM/BW.
Tavella et al. (2021) [64]	*n* = 201 (71.2)	100/101	Secondary Data Analysis of NU-AGE Study	SMI was positively correlated with *Christensenellaceae* R7 and *Ruminococcaceae* UCG 014, 002, 005) and negatively associated with *Fusicatenibacter*, *Blautia*, *Subdoligranulum* and *Ruminococcus* 2.
Ticinesi et al. (2020) [36]	*n* = 17 Sarcopenic *n* = 5 (77) Non-Sarcopenic *n* = 12 (71.5)	3/14	Cross-Sectional	Lower RA of *Faecalibacterium prausnitzii*, *Roseburia inulinivorans*, and *Alistipes shahii* in sarcopenic compared with nonSarcopenic individuals. SPPB score and SMM were significantly different between groups. Genes involved in amino acid metabolism, alpha carotene biosynthesis, flavin biosynthesis, and pyruvate fermentation to acetate were significantly depleted in sarcopenic individuals. Conversely, genes involved in glycolysis and glyoxylate bypass were significantly enriched.
Wang et al. (2022) [67]	*n* = 1417 S *n* = 141 (72.2 ± 8.5) nS *n* = 1276 (62.3 ± 8.5) S: Sarcopenic nS: non-Sarcopenic	S: 48.2% F nS: 60.1% F	Secondary Analysis of Xiangya Osteoarthritis Study	β-diversity, but not α-diversity, was significantly associated with sarcopenia. Sarcopenic individuals had greater RA of *Desulfovibrio piger*, *Clostridium symbiosum*, *Hungatella effluvii*, *Bacteroides fluxus*, *Absiella innocuum*, *Coprobacter secundus* and *Clostridium citroniae*. All but *Coprobacter secundus* was positively associated with sarcopenia severity. Phenylalanine, tyrosine, and tryptophan biosynthesis pathways were depleted, while alpha-linolenic acid metabolism, furfural degradation and staurosporine biosynthesis were enriched in sarcopenic Individuals. *D. Piger* was identified as most important bacteria for discriminating between sarcopenia and non-sarcopenic (AUC: 0.852).
Wang et al. (2023) [68]	*n* = 100 (65–75) Case *n* = 50 (68.4 ± 3.5) Control *n* = 50 (68.7 ± 3.4) Case: Sarcopenia Control: Healthy Controls	0/100	Case–Control	RA of firmicutes and the firmicutes/bacteroidetes ratio was higher in sarcopenic individuals. RA of *Prevotella copri* and *Bifidobacterium longum* were significantly different between groups. RA of *Bacteroides fluxus*, *Bacteroides coprophilus*, *Bacteroides coprocola*, *Bacteroidales bacterium*, *Bacteroides massiliensis*, *Barnesiella intestinihominis*, and *Mitsuokella multacida* was positively associated with skeletal muscle mass, while the RA of *Eggerthella lenta*, *Collinsella aerofaciens*, and *Subdoligranulum variabile* was negatively correlated. AUC of *Bifidobacterium longum* for predicting sarcopenia in older women was 0.647.
Wu et al. (2022) [69]	*n* = 192 Case *n* = 88 (77) Control *n* = 104 (70) Case: Sarcopenia Control: Healthy Controls	87/105	Case–Control	Gut microbiota diversity was lower in the sarcopenia group. Greater RA of firmicutes, bacteroidetes and prevotella/bacteroidetes ration and lower RA of proteobacteria were evident in sarcopenia. *Coprococcus* was positively associated with sarcopenia, whereas *Lachnospiraceae* had a negative relationship.

Note: * Participant number reflects final completions. ** Included overweight because analysis was adjusted by BMI. Abbreviations: ALM: appendicular lean mass; ASM: appendicular skeletal mass; ASMI: appendicular skeletal muscle index; AUC: area under curve; BMI: body mass index; BW: body weight; HGS: handgrip strength; LBM: lean body mass; LM: low mass; NM: normal mass; PF&S: physical frailty and sarcopenia; RA: relative abundance; RCS: repeat chair stands; SMI: skeletal muscle index; SMM: skeletal muscle mass; SNAQ: Simplified Nutritional Appetite Questionnaire; SPPB: Short Physical Performance Battery.

**Table 3 nutrients-17-00081-t003:** Characteristics and main findings of interventional studies.

Study	Participants and Sample Size Per Group *	Age (mean ± SD)	Intervention (Once Daily Dosing)	Duration	Main Outcomes (Intervention vs. Control)
Buigues et al. [52]	15 Males and 35 Females TRE: 28 (9M, 19F) PLA: 22 (6M, 16F)	73.8 ± 1.6 * TRE: 74.2 ± 1.6 PLA: 73.4 ± 1.8 * Average for 50 Enrolled	TRE (Darmocare Pre^®^): Inulin Min. 3375 mg and FOS Min. 3488 mg (per 7.5 g) PLA: Maltodextrin	13 Weeks	TRE vs. PLA Groups: ↑ strength (HGS): 12.4 ± 3.2 vs. 10.2 ± 4.1, *p* = 0.04; ↑ physical performance (self-reported exhaustion ccore): 0.8 ± 1.4 vs. 1.7 ± 1.2, *p* = 0.002; improvement in walking speed, however this did not reach significance.
Ford et al. [55]	21 Females (26 ITT)	73.7 ± 5.6	PRO Supplement: *Bifidobacterium Bifidum* HA-132 (1.54 × 10^9^), *Bifidobacterium Breve* HA-129 (4.62 × 10^9^), *Bifidobacterium Longum* HA-135 (4.62 × 10^9^), *Lactobacillus Acidophilus* HA-122 (4.62 × 10^9^), and *Lactobacillus Plantarum* HA-119 (4.62 × 10^9^). PRE Supplement: 5.6 g Inulin SYN: PRO + PRE Supplements PLA: Maltodextrin	18 Weeks	FFM ↑ from 41.1 ± 1.1 kg at baseline to 43.2 ± 1.2 kg at study end (*p* = 0.03). Carryover effects were seen for *L. Plantarum*, *B Bifidum and L Acidophilus*, indicating a 2-week washout period was not wdequate.
Lee et al. [54]	17 Males and 16 Females PLA: 17 (9M, 8F) TWK10-L: 12 (8M, 4F) TWK10-H: 13 (8M, 5F) TWK10-L: Low-Dose Group TWK10-H: High-Dose Group	PLA: 75.2 ± 7.2 TWK10-L: 77.8 ± 7.2 TWK10-H: 80.5 ± 9.4	TWK10 (*Lactobacillus Plantarum*, Isolated from Taiwanese Pickled Cabbage) Low: 2 × 10^10^ CFU High: 6 × 10^10^ CFU PLA: Maltodextrin	18 Weeks	HGS of the TWK10-H Group at 18 weeks was significantly ↑ (*p* = 0.0187) from baseline. Both TWK10-L and -H Groups demonstrated significantly improved within-Group RCS times at 18 weeks (*p* = 0.0004 and *p* = 0.0008, respectively). Relative MM in the TWK10-H Group was significantly ↑ (*p* = 0.0001) at 18 weeks.
Liu et al. [56]	16 Males and 50 Females uroA: 33 (6M, 27F) PLA: 33 (10M, 23F) 25 and 25 per Protocol Analysis 30 and 30 ITT	71.7 ± 4.9 * uroA: 72.5 ± 5.2 PLA: 71.0 ± 4.6 * Average for 66 Enrolled	Urolithin A 1000 mg PLA: No Details	4 Months	Non-significant improvement in 6MWD in uroA Group (60.8 m, 14.9% change from baseline) vs. PLA Group (42.5 m, 10.1% change from baseline). No significant effect on change in maximal ATP production in *First Dorsal Interosseus* or *Tibialis Anterior*, but ↑ muscle endurance (repeated isometric contractions until fatigue) in uroA vs. PLA.
Neto et al. [53]	SYN: 9 (1M, 8F) PLA: 8 (3M, 5F)	67.9 ± 4.5	6 g FOS, 10^8^ to 10^9^ CFU each of *Lactobacillus paracasei*, *Lactobacillus rhamnosus*, *Lactobacillus acidophilus* and *Bifidobacterium lactis* PLA: maltodextrin	3 Months	No difference (pre/post) in FFM (kg) between SYN and PLA groups (40.2 ± 6.5 vs. 44.4 ± 11.7). No difference (pre/post) in HGS (N) between SYN and PLA groups: (17.2 ± 3.9 vs. 15.7 ± 5.3).
Rondanelli et al. [57]	(Biological Sex was Not Reported) EXP: *n* = 22 PLA: *n* = 28	79.7 ± 4.8 * EXP: 78.8 ± 5.8 PLA: 80.5 ± 3.7 * Average for 50 Enrolled	30 × 10^9^ CFU Lactobacillus Paracasei PS23 (LPPS23), 2.5 g Leucine, and 500 mg Omega-3 Fatty Acids (64.71% EPA, 29.41% DHA, and Remaining 5.88% ω-3 in General) PLA: Isocaloric Placebo	2 Months	Significant improvements in SPPB (+2.22 points), HGS (+4.09 kg) and Tinetti Scale (+2.39), all *p* < 0.05 in EXP Group. Similarly, ALM ↑ significantly (*p* < 0.05) and VAT ↓ significantly (*p* = 0.001) in the EXP compared to the PLA Group over 60 days.

Note: * sample size represents the number of participants who completed the study or for whom complete datasets are available. Arrows denote an increase (↑) or decrease (↓). Abbreviations: 6MWD: 6 min walking distance; ALM: appendicular lean mass; ATP: adenosine triphosphate; CFU: colony forming unit; DHA: docosahexaenoic acid; EPA: eicosapentaenoic acid; EXP: experimental formula, FOS: fructooligosaccharides; FFM: fat-free mass; HGS: handgrip strength; MM: muscle mass; PLA: placebo control; PRE: prebiotic; PRO: probiotic; RCS: repeat chair stands; SPPB: Short Physical Performance Battery; SYN: synbiotic; TRE: Treatment; uroA: urolithin A; VAT: visceral adipose tissue.

**Table 4 nutrients-17-00081-t004:** Evaluation methods for muscle strength, mass and physical performance.

Study	Methods for Evaluation of			
	Muscle Strength	Muscle Mass		Physical Performance
Barger et al. 2020 [58]	HGS	%WBFM	DXA	SPPB
Buigues et al. 2016 [52]	HGS	/	/	GS (4.6 m)
Cox et al. 2021 [59]	RCS	ALM/height2	DXA	/
Dillon et al. 2021 [60]	HGS, RCS (10 Reps)	LBM/height2, ALM/height2	DXA	SPPB, 400 mW, SCPT
Ford et al. 2020 [55]	HGS	BC, FM, FFM and BF (%)	BIA	/
Han et al. 2022 [65]	HGS	SMI, FM	BIA	GS (4 m)
Kang et al. 2021 [61]	HGS, RCS	ASMI	BIA	/
Lee et al. 2021 [54]	HGS, RCS	FM and FFM, Relative Muscle and Fat (%)	DXA	TUG, GS (10 m)
Lim et al. 2021 [62]	HGS	/	/	/
Liu et al. 2022 [56]	MVC of FDI and TA	CSA of FDI and TA	MRI	6MWD
Neto et al. 2013 [53]	HGS	FM and FFM	BIA	/
Picca et al. 2019 [63]	/	ALM, ALM/BMI	DXA	SPPB
Rondanelli et al. 2022 [57]	HGS	FM, FFM, ALM, ALM/BMI and VAT	DXA	SPPB
Shah et al. 2023 [66]	HGS	/	/	/
Sugimura et al. 2022 [70]	/	ASM, BW and SMI (ASM/BW)	BIA	/
Tavella et al. 2021 [64]	HGS	SMI, VAT and SAT, FM, FMI, LM, LMI	DXA	6MWD
Ticinesi et al. 2020 [36]	HGS, RCS	SMI	BIA	SPPB, GS (4 m)
Wang et al. (2022) [67]	HGS, RCS	ASMM	BIA	SPPB, GS (4 m)
Wang et al. 2023 [68]	HGS	BF (%), Limb and Trunk FFM, SMI	BIA	GS (Not Specified)
Wu et al. 2022 [69]	HGS	ASMM	BIA	SPPB

Abbreviations: 400 mW: 400 m walk; 6MWD: 6 min walking distance; ALM: appendicular lean mass; ASMI: appendicular skeletal muscle index; ASMM: appendicular skeletal [muscle] mass; BC: body composition; BF: body fat; BIA: bioelectric impedance analysis; BMI: body mass index; CSA: cross-sectional area; DXA: dual-energy X-ray absorptiometry; FM: fat mass; FFM: fat-free mass; FMI: fat mass index; FDI: first dorsal interosseus; GS: gait speed; HGS: handgrip strength; LBM: lean body mass; LM: lean mass; LMI: lean mass index; MRI: magnetic resonance imaging; MVC: maximum voluntary contraction; RCS: repeat chair stands; SAT: subcutaneous adipose tissue; SCPT: stair climb power test; SMI: skeletal muscle index; SPPB: Short Physical Performance Battery; TA: tibialis anterior; TUG: timed up and go; VAT: visceral adipose tissue; WBFM: whole body fat mass.

**Table 5 nutrients-17-00081-t005:** Evaluation methods for faecal microbiota and microbiome.

Study	Methods for Evaluation of	
	Faecal Microbiota	Faecal Microbiome
Barger et al. (2020) [58]	Faecal Microbiota 16S rRNA Sequencing (V4 Region); Alpha Diversity (ACE, Chao1, Fisher, Shannon, Simpson); Beta Diversity (Unweighted UniFrac, Bray–Curtis, Weighted UniFrac, PCoA)	/
Buigues et al. (2016) [52]	/	/
Cox et al. (2021) [59]	Faecal Microbiota 16S rRNA Sequencing (V4 region); Alpha Diversity (Shannon, Observed Species, Inverse Simpson); Beta Diversity (Weighted UniFrac)	/
Dillon et al. (2021) [60]	Faecal Microbiota 16S rRNA Sequencing (V3-V4 region); Alpha Diversity (Fisher)	Untargeted faecal SCFA metabolomics (GC-MS)
Ford et al. (2020) [55]	Faecal Microbiota 16S rRNA Sequencing (V4 Region)	/
Han et al. (2022) [65]	Faecal Microbiota 16S rRNA Sequencing (V3–V4 region); Alpha Diversity (Shannon); Beta Diversity (Bray–Curtis)	Faecal SCFA metabolomics (GC-MS)
Kang et al. (2021) [61]	Faecal Microbiota 16S rRNA Sequencing (V3–V4 region); Alpha Diversity (Observed, Chao1); Beta Diversity (LEfSe)	PICRUSt
Lee et al. (2021) [54]	/	/
Lim et al. (2021) [62]	Faecal Microbiota 16S rRNA Sequencing (V3–V4 Region); Alpha Diversity (Shannon’s, Pielou’s, Observed, Faith’s; Beta Diversity (Bray–Curtis)	PICRUSt
Liu et al. (2022) [56]	/	Targeted plasma metabolomics (GC-MS and LC-MS)
Neto et al. (2013) [53]	/	/
Picca et al. (2019) [63]	Faecal Microbiota 16S rRNA Sequencing (V3–V4 region); Alpha Diversity (Chao 1 Index)	/
Rondanelli et al. (2022) [57]	/	/
Shah et al. (2023) [66]	Faecal Microbiota 16S rRNA Sequencing (V3–V4 region); Alpha Diversity (Shannon); Beta Diversity (Bray–Curtis)	/
Sugimura et al. (2022) [70]	Faecal Microbiota 16S rRNA Sequencing (V3–V4 Region)	/
Tavella et al. (2021) [64]	Faecal Microbiota 16S rRNA SEQUENCING (V3–V4 region); Alpha Diversity (Chao 1, Observed ASVs, Fisher); Beta diversity (UniFrac, PCoA)	Untargeted serum metabolomics (UPLC)
Ticinesi et al. (2020) [36]	Beta Diversity (Bray–Curtis, PCoA)	Shotgun faecal metagenomics
Wang et al. (2022) [67]	Alpha Diversity (Shannon Index); Beta Diversity (Bray–Curtis)	Shotgun faecal metagenomics
Wang et al. (2023) [68]	Beta Diversity (LEfSe)	Faecal DNA metagenomics
Wu et al. (2022) [69]	Faecal Microbiota 16S rRNA Sequencing (V3–V4 Region); Alpha Diversity (Chao 1 Index and Observed OTU Value)	/

Abbreviations: ACE: abundance-based coverage estimation; ASV: amplicon sequence variance; DNA: deoxyribonucleic acid; GC-MS: gas chromatography–mass spectrometry; LEfSe: linear discriminate analysis effect size; LC-MS: liquid chromatography–mass spectrometry; OTU: operational taxonomic unit; PICRUS: phylogenetic investigation of communities by reconstruction of unobserved states; PCoA: principal coordinate analysis; rRNA: ribosomal ribonucleic acid; SCFA: short-chain fatty acid; UPLC: ultra-performance liquid chromatography.

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
