# Peer review of "The Relationship Between Gut Microbiota, Muscle Mass and Physical Function in Older Individuals: A Systematic Review"

_nutrients, 2024, doi:10.3390/nu17010081_

Round 1
Reviewer 1 Report
Comments and Suggestions for Authors
This excellent and outstanding systematic review examines causality between the microbiome and muscularity in the elderly. The paper is structured and comprehensive. This study could even be considered to be an authoritative landmark paper, to some extent. I have some remarks:
Table 2: Could a meta-analysis be made on the observational studies solely?
Table 3: The interventional studies seem to be too small to draw conclusions.
Table 4: Are all of these anthropometric measurements reliable parameters?
Author Response
Reviewer 1
Reviewers’ Comments: Overall: This excellent and outstanding systematic review examines causality between the microbiome and muscularity in the elderly. The paper is structured and comprehensive. This study could even be considered to be an authoritative landmark paper, to some extent. I have some remarks:
Authors’ Response: We would like to thank reviewer 1 for their time in reviewing our manuscript, and for their insightful and relevant comments. We have tried to address all comments (see below). We hope that we have interpreted all comments correctly and that subsequent changes made to the revised manuscript are to the satisfaction of Reviewer 1.
Reviewers’ Comments: Table 2: Could a meta-analysis be made on the observational studies solely?
Authors’ Response: Thank you to Reviewer 1 for their comment. We did consider undertaking a meta-analysis. However, we decided not to include this because of several reasons: Firstly, it is evident that there is high heterogeneity of the results. Additionally, the included studies often did not comprise similar comparators. Finally, this review aimed to elucidate a potential ‘sarcobiome’ signature comprising of multiple gut microbiota taxa. As changes in multiple taxa would potentially be non‐linear, it could complicate meta‐analysis.
Reviewers’ Comments: Table 3: The interventional studies seem to be too small to draw conclusions.
Authors’ Response: Thank you to Reviewer 1 for their comment and we agree. We have stated within the manuscript the following:
- Line 344-346: Intervention studies that targeted the microbiota failed to show benefit on muscle mass, function and/or indices of sarcopenia. However, results should be interpreted with caution due to their limitations or low-study quality.
- 526-529: Drawing substantive conclusions on the effects of prebiotics and/or probiotics on the gut microbiota is limited due to the heterogeneity of the interventions and research outcomes, study limitations and/or low-quality study designs.
Reviewers’ Comments: Table 4: Are all of these anthropometric measurements reliable parameters?
Authors’ Response: Thank you to Reviewer 1 for their comment and we agree, there are differences in possible accuracy and reliability when directly comparing anthropometric assessments. To further acknowledge this, we have added a sentence in the Limitations section, line 490
Reviewer 2 Report
Comments and Suggestions for Authors
The manuscript presents a solid systematic review concerning the relationship between gut microbiota, muscle mass and physical function in older individuals. The search and selection methods are reported in detail, and the PRISMA flow diagram is presented. The study is based on the analysis of 20 studies selected out of 784 records identified in two literature searches. The reasons for the exclusion of individual studies are reported in the Supplementary material. It is worth mentioning that the researchers asked authors of original papers for additional information if necessary. Both large studies and studies involving small numbers of participants are included.
The conclusions are scientifically sound and interesting. The analysis of relevant studies provides further evidence that the development of sarcopenia is likely influenced by an altered gut microbiome. The available evidence suggests the presence of a ‘sarcobiome’ within sarcopenic with or without frailty compared to non-sarcopenic, healthy older adults.
Remarks:
Perhaps the “sarcobiome” could be briefly recalled in the Conclusions.
Line 26: „69.0 ± 6.4”, please indicate that these values concern age.
Table 2 footnote: please explain SMM, LM, NM, PF&S.
Similarly, please explain all acronyms used in Tables 3 and 4 (they are explained in the text but later on).
References should be formatted strictly according to the MDPI requirements (periods after abbreviations in the journal titles, no capital letters in the titles of articles (except for the beginning).
Author Response
Reviewer 2
Reviewers’ Comments: Overall: The manuscript presents a solid systematic review concerning the relationship between gut microbiota, muscle mass and physical function in older individuals. The search and selection methods are reported in detail, and the PRISMA flow diagram is presented. The study is based on the analysis of 20 studies selected out of 784 records identified in two literature searches. The reasons for the exclusion of individual studies are reported in the Supplementary material. It is worth mentioning that the researchers asked authors of original papers for additional information if necessary. Both large studies and studies involving small numbers of participants are included. The conclusions are scientifically sound and interesting. The analysis of relevant studies provides further evidence that the development of sarcopenia is likely influenced by an altered gut microbiome. The available evidence suggests the presence of a ‘sarcobiome’ within sarcopenic with or without frailty compared to non-sarcopenic, healthy older adults.
Authors’ Response: We would like to thank reviewer 2 for their time in reviewing our manuscript, and for their insightful and relevant comments. We have tried to address all comments (see below). We hope that we have interpreted all comments correctly and that subsequent changes made to the revised manuscript are to the satisfaction of Reviewer 2.
Reviewers’ Comments: Perhaps the “sarcobiome” could be briefly recalled in the Conclusions.
Authors’ Response: Thank you to Reviewer 2 for their comment. We do mention ‘sarcobiome’ again in line 525 and 530 within the conclusions. We hope this is what Reviewer 2 is referring too.
Reviewers’ Comments: Line 26: „69.0 ± 6.4”, please indicate that these values concern age.
Authors’ Response: Thank you to Reviewer 2 for highlighting this oversight, it has now been rectified in Line 206
Reviewers’ Comments: Table 2 footnote: please explain SMM, LM, NM, PF&S.
Authors’ Response: Thank you to Reviewer 2 for highlighting this oversight, it has now been rectified in Line 217-219
Reviewers’ Comments: Similarly, please explain all acronyms used in Tables 3 and 4 (they are explained in the text but later on).
Authors’ Response: Thank you to Reviewer 2 for highlighting this oversight, it has now been rectified in Line 231-234 and 259-265
Reviewers’ Comments: References should be formatted strictly according to the MDPI requirements (periods after abbreviations in the journal titles, no capital letters in the titles of articles (except for the beginning).
Authors’ Response: Thank you for the comment, we have checked our formatting and have used the Nutrients endnote template to format our paper references. We believe we have followed the correct process.